# Memorization or Interpolation? Detecting LLM Memorization through Input Perturbation Analysis

## Abstract

While LLMs achieve remarkable performance through training on massive datasets, they often exhibit concerning behaviors such as verbatim reproduction of training data rather than true generalization. This memorization phenomenon raises critical concerns regarding data privacy, intellectual property rights, and the reliability of model evaluations. This paper introduces PEARL, a novel approach for detecting memorization in LLMs. PEARL assesses the sensitivity of an LLM's performance to input perturbations, enabling detection of memorization in a black-box setting without requiring access to the model's internal parameters or architecture. Furthermore, we investigate how input perturbations affect output consistency, allowing us to distinguish between true generalization and memorization. Extensive experiments on Pythia models demonstrate that our framework robustly identifies cases where models regurgitate learned information. Applied to GPT-4o, PEARL not only detected memorization of classic texts (e.g., the Bible) and common code snippets from HumanEval but also provided evidence suggesting that certain data sources, such as New York Times articles, were likely included in the model's training set. Our anonymized code is available: https://anonymous.4open.science/r/PEARL-2E13/.

## 1 Introduction

The capabilities of large language models (LLMs) are subject to ongoing debate and research within the AI community. A key question is whether their weights encode sensitive personal data, copyrighted material, or proprietary information, a concern that has already led to multiple ongoing legal proceedings. Models can contain sensitive information in two ways: through interpolation or memorization. Studies show that memorization is particularly prevalent in practice (Carlini et al., 2021; Hartmann et al., 2023b).

Recent work has shown that memorization in LLMs is not only common but also scales with model size, data duplication, and prompt length (Carlini et al., 2023; Zhang et al., 2023; Schwarzschild et al., 2024). However, these findings typically rely on access to the model's training data, which is rarely available in practice. As observed by Carlini et al. (2023), current memorization detection techniques face fundamental limitations when the training data is unknown, since they often depend on prompting the model with exact sequences from the training set to reveal memorized content. This limitation is particularly problematic in commercial settings, where training datasets are typically undisclosed and models are accessed via black-box APIs. Major AI providers cite strategic, legal, and security concerns for this lack of transparency (Zewe, 2024; European Parliament, 2023). As a result, there is a pressing need for memorization detection methods that are effective without requiring access to the training corpus.

Despite these limitations, detecting memorization in LLMs, particularly in commercial models, is vital for several interconnected reasons. Without robust memorization detection, organizations face risks, such as privacy breaches, legal non-compliance (Grynbaum & Mac, 2023), and compromised model performance; this is because memorization can blur the line between true learning and mere regurgitation of training data, ultimately eroding public trust in AI systems. From a privacy and security standpoint, such detection is crucial to identify potential leaks of sensitive personal data (Yan et al., 2024; Carlini et al., 2021; Lukas et al., 2023; Yao et al., 2024). Perhaps most importantly,

it fosters transparency and accountability in AI systems by enabling honest communication about model capabilities and limitations (Zhou et al., 2023).

**Relation to prior work.** Early memorization detection methods relied on white-box access, often identifying memorized content via high-confidence generations (Zhou et al., 2023). To address this limitation, black-box methods have been developed. For instance, Carlini et al. (2023) introduced a prompting-based approach to detect exact suffix reproduction, while Zhang et al. (2023) proposed the concept of counterfactual memorization, which measures prediction shifts when training examples are removed. More recent black-box techniques include analyzing output distributions across repeated prompts (Zhou et al., 2024), crafting detection prompts to elicit memorized responses (Golchin & Surdeanu, 2024), and quantifying memorization through the minimal prompt length required to elicit it (Schwarzschild et al., 2024). Despite their effectiveness, these techniques often require access to training data and tend to conflate membership inference with memorization, limiting their real-world applicability. A key open challenge, therefore, is distinguishing between ***true memorization***, where the model's performance is fragile around a specific data point, and ***successful interpolation***, where the model generalizes from similar examples. A detailed related work section is in Appendix A.2.

This leads to two fundamental questions: ❶ *How can we detect memorization when the training data is unavailable?* and ❷ *How can we do so in a black-box setting without access to model internals?*

**This paper.** To address these research questions, we propose the following ***Input Perturbation Sensitivity Hypothesis*** (IPSH):

> For a given *model*, *task*, and *data point*, if the model has memorized the data point, then its task performance will be sensitive to perturbations of that data point.

Building on this hypothesis, we propose **PEARL** (**PE**rturbation **A**nalysis for **R**evealing **L**anguage model memorization, a black-box framework that detects memorization by measuring a model's sensitivity to controlled perturbations.

**Contributions.** Our main contributions are: (i) We introduce and motivate the IPSH, establishing its connection to memorization in LLMs. (ii) We propose PEARL, a practical black-box framework that leverages IPSH to detect memorization without requiring access to training data or model internals. (iii) We empirically validate PEARL on the open-source Pythia model suite, demonstrating its efficacy in detecting memorization using only input-output behavior. (iv) We apply PEARL on the commercial model GPT-4o, uncovering instances of memorization across diverse domain, including canonical texts (e.g., the Bible), benchmark code (e.g., HumanEval), and proprietary content (e.g., New York Times articles), illustrating its practical utility in real-world settings.

## 2 THE MEMORIZATION ADVANTAGE AND IPSH

LLMs are pretrained on massive datasets that include a significant portion of publicly available digital text up to their training cutoff date. As these models are continually updated, their training corpora are refreshed to include more recent content. This practice, however, introduces a critical challenge for evaluation: the risk of data contamination (Zhu et al., 2024; Dong et al., 2024), where test benchmarks overlap with the training data. When such overlap occurs, models can achieve high performance by recalling memorized examples rather than by demonstrating robust generalization, thereby inflating standard performance metrics.

This issue is exacerbated when a large portion of the test set is present in the training data. Critically, performance gains attributable to memorization are fragile and non-transferable, failing to represent the model's capabilities in real-world deployment scenarios where exact training matches are rare. We formally define the *memorization advantage* as the inflated performance on a test set that a model gains by relying on memorized content, as opposed to genuine generalization.

Let $\mathcal{D}_{\text{train}}$ be the pretraining data of an LLM, $\mathcal{D}_{\text{test}}$ be the test data for some task $b$, and $\mathcal{D}'_{\text{test}}$ be fresh testing data for $b$, where $\mathcal{D}_{\text{train}} \cap \mathcal{D}_{\text{test}} \neq \emptyset$ (*i.e.* the test data is contaminated) and $(\mathcal{D}_{\text{train}} \cup \mathcal{D}_{\text{test}}) \cap \mathcal{D}'_{\text{test}} = \emptyset$, (*i.e.* $\mathcal{D}'_{\text{test}}$ is uncontaminated). Let $m_b$ be a performance measure for the task $b$. Let $\theta^{\bullet}$ be a model that has memorised $\mathcal{D}_{\text{train}} \cup \mathcal{D}_{\text{test}}$. Let $\theta^{\circ}$ be a model that may have been trained on $\mathcal{D}_{\text{train}}$, but did not memorize any of its data instance in $\mathcal{D}_{\text{train}}$. The memorization advantage of $\theta^{\bullet}$ over $\theta^{\circ}$ is

Table 1: Illustration of memorization and interpolation for a text completion task. The highlighted text indicates the modification applied based on IPSH to the original input. While multiple valid completions are possible, the model's output for the memorized instance exhibits incoherence. In contrast, the completion for the generalized instance remains coherent, even after the perturbation is introduced.

| Input | Output |
|---|---|
| **Memorization** | |
| Complete the following text: Hagrid raised a gigantic fist and knocked three times on the castle door. | Hagrid raised a gigantic fist and knocked three times on the castle door. **The sound echoed through the ancient stone walls, reverberating like a drumbeat in the silent night.** |
| Complete the following text: Hagrid graised a giganticfist and knocked [] times on the door. | Hagrid graised a giganticfist and knocked [] times on the door. **Step 1: Corrected the spelling errors in the given text. Step 2: Added a number to complete the sentence.** |
| **Generalization** | |
| Complete the following text: Albert Einstein was a theoretical physicist who is known for developing the theory of relativity, | Albert Einstein was a theoretical physicist who is known for developing the theory of relativity, **which revolutionized our understanding of space, time, and gravity.** |
| Complete the following text: AlbertEinstein was a theoretical [] who is known for developing the theory of relative. | AlbertEinstein was a theoretical [] who is known for developing the theory of relative. **This groundbreaking theory, which includes both the Special Theory of Relativity and the General Theory of Relativity, revolutionized our understanding of space, time, and gravity.** |

$$ma(\theta^{\bullet}, \theta^{\circ}) = \mathbb{E}_{x \sim \mathcal{D}_{\text{test}}}[m_b(\theta^{\bullet}(x)) - m_b(\theta^{\circ}(x))] - \mathbb{E}_{y \sim \mathcal{D}_{\text{test}'}}[m_b(\theta^{\bullet}(y)) - m_b(\theta^{\circ}(y))] \quad (1)$$

This formulation isolates the advantage gained specifically from having seen the test instances by subtracting the general performance difference between the two models.

**Memorization vs. Membership Inference.** It is crucial to distinguish memorization from membership. A data point can be a member of the training set without being memorized (e.g., if the model generalizes from it). Conversely, a data point can only be memorized if it is a member of the training set. Therefore, memorization identifies a subset of the training data members. In contrast, membership inference seeks to determine whether a given data point was in the training set, regardless of whether it was memorized (Shokri et al., 2017; Duan et al., 2024). In Appendix A.10, we also compare memorization detected by PEARL and membership inference.

**IPSH: Input Perturbation Sensitivity Hypothesis.** A central challenge in assessing LLMs is distinguishing verbatim memorization of training data (Carlini et al., 2021; Yao et al., 2024; Hartmann et al., 2023b) from robust interpolation. While some argue memorization may precede generalization (Feldman, 2020; Dankers & Titov, 2024), it ultimately signifies fragile performance, a key indicator of overfitting (Schwarzschild et al., 2024; Duan et al., 2024). Our hypothesis offers a path to disentangle these concepts through perturbation sensitivity. Table 1 illustrates this distinction with real-world examples from the amazon-nova-lite-v1.0 model. When prompted to complete excerpts from Harry Potter and Wikipedia, the model produces the correct output. This raises an important question: "*How can we determine whether a model is truly interpolating or simply reproducing a memorized data point?* "

We hypothesize that the answer lies in the model's robustness to input perturbations. This simple experiment provides initial support for IPSH: perturbing memorized content (e.g., the Harry Potter text) disrupts model performance, while the model generalizes robustly for non-memorized inputs (e.g., the Einstein Wikipedia text).

In Figure 1, we show the result of another preliminary experiment, using GPT-4o on the completion task, using a Shakespearean poem (confirmed as training data) and a news article published after the model's release (See Appendix A.5). Here, we automated perturbation by randomly flipping bits in the input text (See Section 3) and measured performance using the normalized compression distance (NCD). For the Shakespeare sample, we observe a substantial performance drop of 2% vis-a-vis the

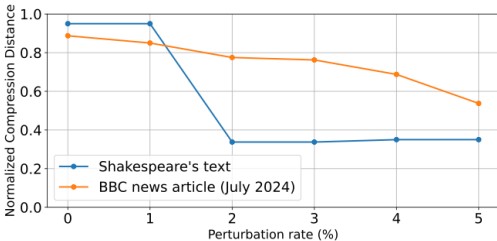

Figure 1: Performance sensitivity of GPT-4o to bit-flip perturbation. The memorized Shakespeare poem shows a significant performance drop at 2% perturbation, while the non-memorized BBC article degrades gradually.

BBC article published after GPT-4o was trained. This result suggests that we can build a classifier to identify memorized instances.

The observed asymmetry — degradation vs. resilience — in both experiments aligns with the idea that memorization relies on verbatim recall.

**An Operational Measure of Memorization.** To quantify the sensitivity of a model's performance to perturbing memorized instances, we introduce formal concepts that underpin our detection method, PEARL. Formally, given a task $b$, a performance measure for that task $m_b$, and a data point $x$, the $\epsilon$-performance neighborhood function for $x$ is

$$N_b(x) = \{x_i\}_{i=1}^k \text{ s.t. } \left| m_b(x) - m_b(x_i) \right| < \tau_b. \tag{2}$$

Here, $N_b$ generates $k$ distinct inputs that are semantically similar under the measure $m_b$. $N_b$ is declarative: it does specify how to construct neighborhoods. It needs to be instantiated for each task; Section 3 describes how we have instantiated $N_b$ in this work.

To measure memorization as performance fragility, we propose Generalization Failure Density (GFD), which quantifies how often a model $M$'s performance drops sharply in the neighborhood of a point:

$$\text{GFD}(M, x, m_b, N_b) = \frac{1}{k} \sum_{y_i \in N_b(x)} \mathbf{1} \left[ m_b(M(x)) - m_b(M(y_i)) > \tau_b \right] \tag{3}$$

where $M$ is the model and $y_i \in N_b(x)$. The proposed metric measure captures the model's sensitivity to small input changes. What counts as "small", denoted by $\tau_b$, is task-specific: in some domains, even a minor syntactic edit, like adding a logical not at the root of a Boolean expression, can yield a major semantic shift, and both the model and GFD should both be sensitive to such cases. The key is, however, to define $N_b$ so that it emphasizes syntactic variations around $x$ that largely preserve semantics since, under IPSH, if the model has memorized $x$, its performance should be syntactically brittle in $x$' neighborhood. A low GFD indicates robust generalization around $x$, while a high GFD suggests localized generalization failures, consistent with over-reliance on specific patterns (*i.e.* memorization). Which values of GFD indicate memorization or not is task-dependent and requires another threshold. GFD averages the neighborhood because even models that perfectly interpolate will be susceptible to adversarial examples in their neighborhood Szegedy et al. (2013).

## 3 PEARL: PERTURBATION ANALYSIS FOR REVEALING LANGUAGE MODEL MEMORIZATION

We design PEARL as a novel framework that builds upon our IPSH, which posits that a model's task performance is sensitive to perturbing memorized data points. By systematically implementing this hypothesis through controlled input variations, PEARL provides a robust method to detect memorization in LLMs. As illustrated in Figure 2, the framework quantifies performance variation in the neighborhood of a data point (See Equation (3)), enabling the identification of memorization and generalization patterns.

**Neighbor data point generation:** The first step in PEARL is instantiating $N_b$, in order to generate neighbors on an instance $x$. We do so via transformation. Specifically, we use bit flips, which are

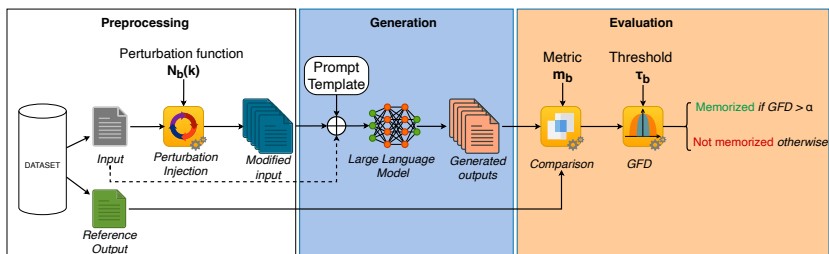

Figure 2: Overview of the PEARL framework for identifying memorization in LLMs based on IPSH.

quite low level, but are task universal. We cannot guarantee that the transformed instance is within $\tau_b$ of a transformed instance, but empirically we have not observed violations of this constraint. Our transformation perturbs the input $x$ with intensity $k$, controlling its deviation from the original. We use two perturbations: (i) flipping $k\%$ of bits in the text's binary form and decoding back, and (ii) replacing $k$ words with synonyms from a predefined dictionary. We set $k \in \{0, \ldots, 5\}$ to create similar inputs while preserving semantics. These perturbations satisfy Equation (2) by producing task-consistent variants: bit flips introduce minimal changes, while synonym substitution follows Firth's dictum, assuming that meaning — and thus performance — remains stable under synonym replacement for tasks like completion and summarization. In Appendix A.9, we also analyze the impact of synonym substitution as an alternative transformation to highlight its effect on sensitivity. Converting GFD into a classifier varies by task, but a single threshold worked in our experiments, so, in this work, we considered instances with GFD $> 0.2$ to be memorized.

## 4 EXPERIMENTAL SETUP

We evaluate PEARL to validate IPSH through two sets of experiments. The first involves open-source models with known training data. We fine-tune on our positive test set for multiple epochs to ensure that the data tested for memorization has indeed been seen by the model. The second involves a closed-source model to develop real-world case studies, demonstrating PEARL's potential for detecting memorization.

**Model Selection.** We evaluate PEARL using both open- and closed-source LLMs. For open models, we consider the Pythia suite (Biderman et al., 2023), which includes models ranging from 70M to 13B parameters. We focus on versions trained on the deduplicated variant of it training dataset, Pile (Gao et al., 2020) to reduce memorization due to repeated pattern. This setup allows us to analyze memorization under controlled conditions.

To validate our approach in real-world scenarios, we extended our analysis to GPT-4o, examining memorization detection in a black-box context where model internals parameters are inaccessible. The selection of GPT-4o is particularly relevant given recent developments in AI ethics and intellectual property rights. Notably, in light of the January 2024 legal dispute between OpenAI and The New York Times regarding alleged copyright infringement and unauthorized use of journalistic content for training purposes (Grynbaum & Mac, 2023). We thus proposed to investigate potential memorization of New York Times articles in GPT-4o.

**Datasets.** To evaluate PEARL, we use datasets for both open- and closed-source models. For Pythia, we create two sets: a positive set of 1,000 samples from The Pile (Gao et al., 2020), fine-tuned over multiple epochs to induce memorization, and a negative set of 1,000 samples from RefinedWeb (Penedo et al., 2023), disjoint from The Pile to favor generalization. For GPT-4o, we use four 100-sample sets: HumanEval (Chen et al., 2021), often suspected in training (Dong et al., 2024); LBPP, released post-training (Matton et al., 2024); Bible passages, likely memorized; and New York Times articles, given legal scrutiny (Grynbaum & Mac, 2023). These cover varying memorization likelihoods and domains.

**Tasks & Evaluation Metrics.** We consider four common tasks in Natural language processing and Software engineering: text completion and text summarization as well code completion and code summarization. We distinguish the tasks because in code summarization, the input is code written in

a programming language while the output is text written in natural language. We use different metrics for model performance evaluation depending on the tasks. For code/text completion, we rely on the Normalized Compression Distance (NCD). For code/text summarization, we use ROUGE-L (Lin, 2004). To obtain reliable estimates, we sample the model $i = 10$ times for each perturbation intensity $k$ and use the average performance to compute task-level metrics.

# 5 EXPERIMENTAL RESULTS

In this section, we empirically evaluate the PEARL framework to validate the Input Perturbation Sensitivity Hypothesis (IPSH) in detecting memorization across open- and closed-source LLMs. Specifically, our experiments are designed to answer the following research questions:

**RQ1.** *How effectively does* PEARL*'s perturbation analysis identify memorization in LLMs?*
**RQ2.** *How accurately does* PEARL *detect memorized instances when applied to commercial models like GPT-4o?*
**RQ3.** *How does* PEARL *characterize differences in memorization behavior across tasks?*

**Validation of IPSH.** We first validate IPSH in a controlled setting using the Pythia-410M model. As Pythia is a base generative model not trained for instruction following, we evaluate it on a text completion task. Figure 3 illustrates the text-splitting procedure used to create prompts and reference outputs. For each sample, we generate perturbed inputs, prompt the model repetitively, obtain the outputs, and compute the performance metric i.e. Normalized Compression Distance (NCD), and the Generalization Failure Density (GFD) as defined in Equation 3.

A key step is determining a sensitivity threshold $\tau_b$ for identifying memorized instances. We calibrate this threshold on the RefinedWeb dataset, which is known to be excluded from Pythia's training data. We observed that false positive rates decreased steadily with increasing $\tau_b$, exhibiting a noticeable inflection point around $0.2$. (See Figure 10 in Appendix). We therefore set $\tau_b = 0.2$ for subsequent experiments, as it provides a balanced trade-off between sensitivity and specificity when distinguishing between memorized and non-memorized instances.

Input **X**

Albert Einstein was a theoretical physicist who is known for developing the theory of relativity. This groundbreaking theory, which includes both the Special Theory of Relativity and the General Theory of

Relativity, revolutionized our understanding of space, time, and gravity.

80 %    20 %

Reference Output **Y**

Figure 3: Example input $X$ and reference output $Y$ for a text completion task.

Applying PEARL with $\tau_b = 0.2$, we compare the proportion of of memorized instances identified in the Pile (training data) versus RefinedWeb (non-training data). Figure 4 shows that PEARL identifies an order of magnitude more memorized samples in the Pile ($> 20\%$ at epoch #10) than in RefinedWeb ($\simeq 2\%$ at epoch #10). Furthermore, with a higher sensitivity threshold (e.g., $\tau_b = 0.4$, false positives on RefinedWeb are nearly eliminated, albeit with a stricter identification cutoff for the Pile. For completeness, we also analyze synonym substitution as an alternative transformation and observe a similar trend to those obtained with bit flips. The results further show that as we iterate in the fine-tuning process, the number of detected memorized instances increases for the Pile (training data) while the number of detected instances remains stable and low for RefineWeb (not part of the training data).

> **Answer to RQ1:** PEARL effectively distinguishes between memorized and non-memorized data in a controlled setting, providing strong empirical validation for the IPSH.

**Impact of Model Size.** Prior work from Carlini et al. (2023) has argued that larger models have increased capacity for memorization. We repeat the experiments of RQ1 with different size-variants of Pythia. Our results confirm this phenomenon: as shown in Table 2, the number of memorized instances identified by PEARL increases consistently with the model size, even for high sensitivity thresholds. For instance, at a sensitivity threshold of $\tau_b = 0.2$, PEARL identifies 110 memorized instances in the 410M model, but 259 in the 6.9B model (a 135% increase). Most notably, this scaling effect persists even under the most stringent threshold $\tau_b = 0.2$, where the Pythia 6.9B model identifies 3.4x more memorized instances than the 410M model (110 vs. 32).

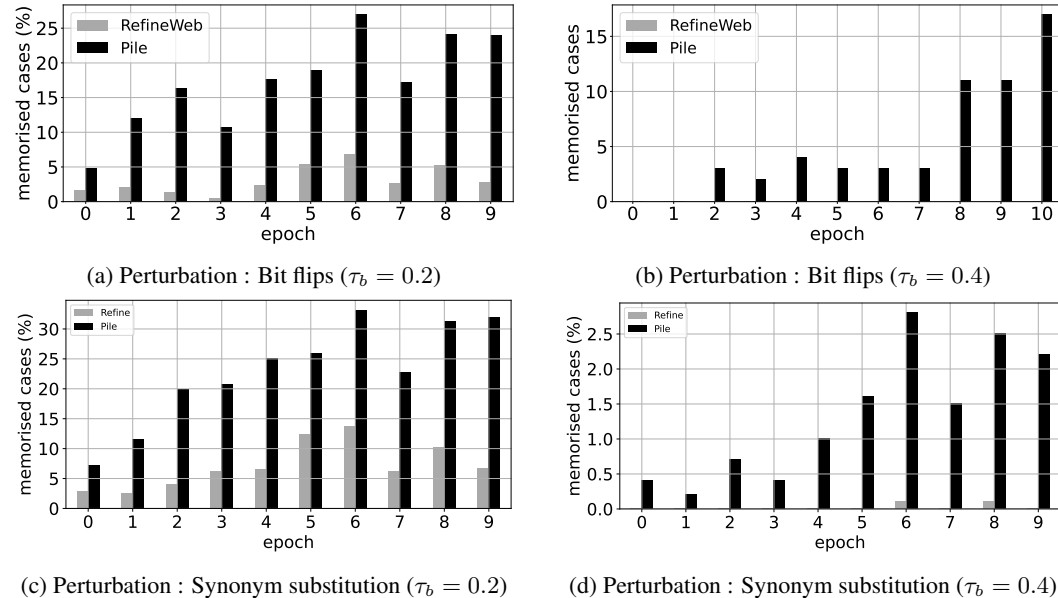

(a) Perturbation : Bit flips ($\tau_b = 0.2$)

(b) Perturbation : Bit flips ($\tau_b = 0.4$)

(c) Perturbation : Synonym substitution ($\tau_b = 0.2$)

(d) Perturbation : Synonym substitution ($\tau_b = 0.4$)

Figure 4: Proportion of memorized instances identified by PEARL in Pythia-410m across datasets and perturbation types at different epochs. The Pile shows high sensitivity to perturbations, unlike RefinedWeb. (a) and (b) use bit flips perturbation , while (c) and (d) use synonym substitution as an alternative perturbation. Examples of both methods are showin in Appendix A.9.

Table 2: Number of memorization cases identified by PEARL on Pythia models of varying sizes, evaluated under different sensitivity thresholds $\tau_b$. The results show an increasing proportion of memorized data with model size, indicating that larger models tend to memorize more.

| Model | #identified memorized instances $\tau_b = 0.16$ | #identified memorized instances $\tau_b = 0.2$ | #identified memorized instances $\tau_b = 0.25$ |
|---|---|---|---|
| Pythia 410m | 265 | 110 | 32 |
| Pythia 1.4B | 349 | 171 | 57 |
| Pythia 6.9B | **449** | **259** | **110** |

**GPT-4o Memorization.** To investigate memorization in commercial models, we apply PEARL to GPT-4o using datasets with varying likelihoods of being memorized. We use either code or text completion tasks as appropriate for each dataset. Given that LBPP is composed of samples that are less likely to have been included in the GPT-4o training set (Matton et al., 2024), we first apply PEARL on this dataset to determine the optimal sensitivity threshold values that minimize the number of false positives (see Figure 10).

Table 3 shows the number of memorization cases that PEARL identifies among each subset of 100 samples within the Bible, HumanEval and NY Times datasets. PEARL identifies a high proportion of

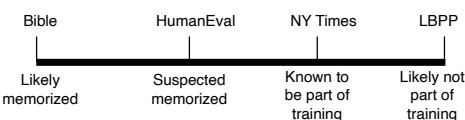

Figure 5: Sample sets from different sources with varying probabilities of being memorized.

| Data source | #identified memorized instances $\tau_b = 0.14$ | #identified memorized instances $\tau_b = 0.24$ | #identified memorized instances $\tau_b = 0.29$ |
|---|---|---|---|
| HumanEval | **60** | **30** | **23** |
| Bible | 42 | 3 | 2 |
| NY Times | 5 | 1 | 0 |

Table 3: GPT-4o memorization counts

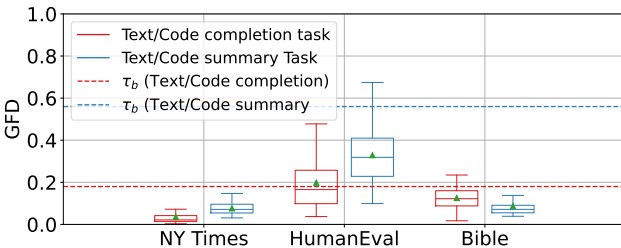

Figure 6: Distribution of sensitivity scores for GPT-4o across datasets and tasks. The dashed horizontal lines represent the threshold calibrated on the LBPP dataset. ❶ The results show that a significant proportion of the Bible and Human Eval data exhibit sensitivity above the established threshold, assuming memorization of some of their instances. ❷ The lower sensitivity distribution of the data on the summary task indicates that the memorization pattern is task-dependent.

memorized instances in the HumanEval code benchmark (60/100 at $\tau_b = 0.14$), indicating extensive memorization of canonical programming problems. Memorization is also present but less prevalent in the Bible (42/100 at $\tau_b = 0.14$), with the effect becoming negligible at higher thresholds. In contrast, PEARL detects minimal memorization of New York Times articles ($\leq 5/100$ across all thresholds).

**Answer to RQ2:** PEARL reveals that GPT-4o exhibits strong memorization of the HumanEval code benchmark, moderate memorization of biblical text, and minimal memorization of New York Times articles.

Appendix A.8 shows examples of the highest and lowest sensitivity scores (GFD) in the Bible and NYT datasets. High-sensitivity (memorized) samples are often stylistic or content outliers, supporting the notion that memorization compensates for a lack of generalizable patterns.

**Task Dependence of PEARL.** We investigate the robustness of PEARL's memorization detection to the choice of task. We evaluate GPT-4o on the same datasets (Bible, HumanEval, NYT) using code/text summarization tasks, with the threshold $\tau_b$ calibrated on LBPP to minimize false positives.

The results demonstrate a strong task dependence. When switching from completion to summarization tasks, only 6 of the previously identified memorized samples were still detected as memorized. Figure 6 shows the distribution of sensitivity values computed for each sample based on the two tasks: with the completion tasks, we can identify memorization cases of GPT-4o with a large number of HumanEval code samples and some Bible verses. There is also a statistically significant difference between the medians of sensitivity values for these Human Eval and NYT. In contrast, with the summarization tasks, only a few samples from HumanEval are identified as memorized, and the difference between datasets is no longer significant. This can be explained by the nature of the summarization task: the input context itself contains the necessary information, allowing GPT-4o to generalize effectively. Therefore, perturbations cause less severe performance degradation compared to completion tasks that may require verbatim recall.

**Answer to RQ3:** PEARL's framework application reveals distinct memorization patterns across different tasks, demonstrating that memorization behavior is task-dependent.

**Memorization is not Membership.** We investigate the relationship between memorization and membership inference by analyzing a series of Pythia-410M checkpoints finetuned on the Pile dataset. While memorization of a data point implies its membership in the training set, the converse is not necessarily true. Our analysis focuses on this asymmetry.

For each training epoch, we performed a membership inference attack (Carlini et al., 2021) by evaluating the perplexity of the dataset described in the evaluation setup section. The objective is to identify data points with low perplexity, a characteristic that indicates the model has assigned a high probability to these inputs. Consistent with our previous experiments, we selected a perplexity threshold of 60, which represented an optimal trade-off, maintaining a low false positive rate (FPR) for effectively classifying potential members. We then compared these membership inference results with our memorization detection outcomes (using a threshold of $\tau_b = 0.2$). As summarized in Table 4, this comparative analysis reveals the evolving relationship between membership inference

| Epoch | Pile | | | RefinedWeb | | |
|---|---|---|---|---|---|---|
| | MIA | Memo | Intersection | MIA | Memo | Intersection |
| 0 | 99.1 | 11.0 | 10.8 | 74.7 | 3.9 | 2.8 |
| 1 | 95.5 | 11.1 | 10.6 | 53.9 | 2.6 | 1.7 |
| 2 | 34.8 | 12.6 | 4.4 | 8.4 | 3.0 | 0.3 |
| 3 | 28.6 | 12.1 | 3.2 | 3.0 | 2.3 | 0.1 |
| 4 | 49.0 | 14.3 | 7.1 | 3.8 | 2.4 | 0.1 |
| 5 | 39.3 | 16.9 | 7.0 | 1.9 | 2.4 | 0.1 |
| 6 | 46.8 | 18.3 | 8.5 | 1.9 | 2.6 | 0.0 |
| 7 | 60.8 | 22.1 | 14.2 | 3.1 | 2.9 | 0.1 |
| 8 | 68.5 | 22.3 | 14.4 | 3.2 | 2.7 | 0.1 |
| 9 | 76.4 | 22.3 | 16.3 | 4.1 | 2.7 | 0.1 |
| 10 | 83.8 | 22.5 | 18.9 | 4.2 | 2.0 | 0.1 |

Table 4: Proportion of data identified as memorized (Memo) and as training members (MIA) based on perplexity (%).

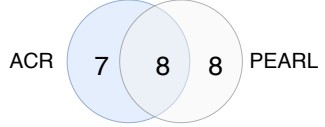

Figure 7: Overlap of detected memorization cases using IPSH and ACR.

and memorization detection across different checkpoints. A key observation is that not all member instances exhibit memorization, highlighting the distinction between these two concepts.

**Adversarial Compression Ratio (ACR) vs PEARL.** We compare PEARL against state-of-the-art white-box ACR method (Schwarzschild et al., 2024) on the Pythia-410M model. Since ACR was originally applied on a subset of 100 samples of FamousQuotes dataset, we execute PEARL with the text completion task on this dataset for a fair comparison. Using the same sensitivity threshold ($\tau_b = 0.2$) calibrated on RefinedWeb, PEARL detects 16 memorized instances, which is comparable to the 15 instances detected by ACR. Furthermore, as shown in Figure 7, there is a 50% overlap between the memorization instances identified by both methods. This result demonstrates that PEARL, despite being a black-box method, achieves detection efficacy on par with a state-of-the-art white-box approach, without requiring access to the model's internal states.

**Theoretical Analysis and Failure Cases of IPSH.** Due to space constraint, we show in Appendix A.1 the theoretical analysis and failure cases of IPSH. The hypothesis formalizes memorization as performance fragility under input perturbations, linking it to sharp minima in the loss landscape. It holds when task metrics align with memorized content and perturbations are informative. Theoretical failure cases include false positives from incomplete generalization and false negatives from robustly memorized patterns. Consequently, the IPSH is an implication, not an equivalence, making PEARL a powerful but calibrated detector of memorization characterized by performance fragility.

**Impact and Limitations.** The validation of IPSH through PEARL provides a principled framework for detecting memorization in LLMs. This enables researchers and practitioners to audit how models store and utilize training data, contributing directly to the ongoing debates around data ownership and consent in AI training. However, PEARL has a limitation: its sensitivity threshold ($\tau_b$) requires calibration using data known to be outside the training set. This requirement, however, does not hinder its applicability, as such data can typically be sourced or constructed in practice, even for closed-source models.

## 6 CONCLUSION

In this paper, we introduce PEARL, a black-box framework that operationalizes the input perturbation sensitivity hypothesis (IPSH) for detecting memorization in LLMs. Through extensive experiments on both open-source (Pythia) and commercial (GPT-4o) models across diverse datasets, we demonstrated that perturbation sensitivity serves as a reliable indicator of memorization, particularly in tasks requiring exact content reproduction. Our findings reveal that memorization patterns are strongly task-dependent and more prevalent in unique or distinctive content. By providing a practical tool for auditing memorization without requiring access to training data or model internals, PEARL contributes to addressing critical challenges in data privacy, model evaluation, and responsible AI development.

**Reproducibility Statement.** To facilitate reproducibility, we provide the full source code for our experiments at https://anonymous.4open.science/r/PEARL-2E13/.

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

# A  APPENDIX

## A.1  THEORETICAL ANALYSIS OF THE IPSH

This section provides a formal framework for the Input Perturbation Sensitivity Hypothesis (IPSH), examining its theoretical underpinnings and boundaries using the notation established in this paper.

### A.1.1  FORMAL DEFINITIONS AND SETUP

Let us define a data point as a sequence $x$, a task $b$ with a performance metric $m_b$, and a model $M$. Let $N_b(x)$ be the neighborhood function generating a set of perturbations $\{x_i\}_{i=1}^k$ of $x$, as defined in Equation (2).

We define the *Performance Sensitivity* $S_M(x, b)$ of model $M$ to point $x$ on task $b$ as the expected drop in performance under perturbations:

$$S_M(x, b) = \mathbb{E}_{x_i \sim N_b(x)} \left[ \max\left(0, m_b(M(x)) - m_b(M(x_i))\right) \right]$$

The IPSH can then be formally stated as:

> **Hypothesis (IPSH):** For a model $M$ trained on dataset $\mathcal{D}_{\text{train}}$, there exists a threshold $\tau_b > 0$ such that for a point $x \in \mathcal{D}_{\text{train}}$,
>
> $$x \text{ is memorized} \implies S_M(x, b) > \tau_b.$$
>
> Conversely, if $x$ is generalized via robust interpolation, then $S_M(x, b) \le \tau_b$.

The Generalization Failure Density (GFD) introduced in Equation (3) serves as an empirical estimator of this sensitivity:

$$\text{GFD}(M, x, m_b, N_b) \approx \Pr_{x_i \sim N_b(x)} \left[ m_b(M(x)) - m_b(M(x_i)) > \tau_b \right]$$

### A.1.2  THEORETICAL CONDITIONS FOR THE IPSH TO HOLD

The IPSH is grounded in the geometry of the loss landscape. It holds most strongly under these theoretical conditions:

- **Condition 1 (Sharp vs. Flat Minima):** Memorization corresponds to $x$ lying in a *sharp minimum* of the loss landscape. Formally, the Hessian $\nabla_\theta^2 \mathcal{L}(x, \theta)$ has large positive eigenvalues for memorized $x$. A perturbation $x_i$ effectively induces a parameter shift $\delta\theta$, causing a large loss increase $\delta\mathcal{L} \approx \frac{1}{2}\delta\theta^T \nabla_\theta^2 \mathcal{L} \delta\theta$ for memorized points, manifesting as a performance drop in $m_b$. Generalized points lie in *flat minima*, where the eigenvalues are small, and are thus robust to such perturbations.

- **Condition 2 (Task Alignment with Memorization Type):** The task $b$ must be such that the performance metric $m_b$ is sensitive to the specific information that was memorized. If $b$ is text/code completion and $x$ was memorized verbatim, then $m_b$ will be high for $x$ and low for $x_i$. If $b$ is summarization, the same memorized information may not be necessary for a high score, violating this condition and leading to a failure of the IPSH for that task—as empirically observed in Section 5.

- **Condition 3 (Informative Perturbations):** The neighborhood function $N_b(x)$ must generate perturbations $x_i$ that are semantically similar but syntactically distinct from $x$. The perturbations should be designed to disrupt the memorized pathway while preserving the task's fundamental requirements, as achieved through the bit-flip and synonym substitution strategies described in Section 3.

### A.1.3  THEORETICAL FAILURE CASES AND LIMITATIONS

The converse of the IPSH is not always true ($S_M(x, b) > \tau_b \not\Rightarrow x$ is memorized). The hypothesis can fail theoretically in these scenarios:

- **Failure Case 1 (Boundary of Competence):** A model may have learned a robust but highly local pattern. If the perturbation $x_i$ falls outside the model's region of competence for that pattern, performance drops. This is a form of *incomplete generalization*, not memorization, but would yield a high GFD score, causing a false positive. This is why calibration on held-out data like RefinedWeb is crucial for setting $\tau_b$.

- **Failure Case 2 (Robust Memorization):** The theory assumes memorization is always brittle. However, if a model memorizes a higher-order *rule* or *template* (e.g., a grammatical structure, a proof strategy) that is itself robust to perturbations, then $S_M(x, b)$ may remain low even for a memorized point. This represents a case where the line between memorization and generalization is blurred.

- **Failure Case 3 (Mismatched Task-Metric Pair):** If the task $b$ and metric $m_b$ are not aligned with the type of memorization (e.g., using ROUGE-L to detect verbatim code memorization), the sensitivity signal may be weak even for truly memorized content.

A.2 DETAILED RELATED WORK

We review existing research on memorization in large language models, with a focus on detection techniques, the distinction between memorization and membership inference, and the limitations of current black-box and white-box approaches.

**Memorization vs. Generalization.** The relationship between memorization and generalization in LLMs is complex and often difficult to disentangle. While generalization is the intended outcome of training, memorization, particularly of rare or sensitive data is undesireable leading to privacy concerns and misleading model performance. Studies such as (Tirumala et al., 2022) and (Kiyomaru et al., 2024) show that memorization increases with model scale, prompt length, and training dynamics, even in the absence of overfitting. This distinction becomes even more blurred in the presence of data contamination. Jiang et al. (2024) demonstrate that even minor overlaps between training and evaluation data can significantly inflate performance, making it difficult to determine whether a model is truly generalizing or merely recalling memorized content.

To address this ambiguity, several works have proposed several definitions of memorization, including verbatim, approximate, and counterfactual memorization (Feldman & Zhang, 2020; Feldman, 2020; Ishihara, 2023; Hartmann et al., 2023a). Recent work has deepened our understanding of memorization mechanisms. (Speicher et al., 2024) characterized learning phases in memorization, while (Dankers & Titov, 2024) demonstrated that memorization occurs gradually across model layers, with early layers playing a more crucial role. In an effort to formalize the boundary between memorization and generalization, (Wang et al., 2025) introduce the concept of distributional memorization. Their findings show that LLMs tend to rely more on memorization in knowledge-intensive tasks, such as factual question answering, while generalization is more prominent in reasoning-oriented tasks. To reduce memorization, (Lee et al., 2022) and (Kandpal et al., 2022) propose deduplication techniques that significantly reduce memorization and improve training efficiency. (Ippolito et al., 2023) introduce MEMFREE decoding to block verbatim generation, though paraphrased leakage persists. These findings motivate the need for robust detection methods that can reliably identify memorization, especially in black-box settings where training data and model internals are inaccessible.

**Training Data Extraction and Membership Inference.** Early work by (Carlini et al., 2021) demonstrated that LLMs can memorize and regurgitate verbatim training data, even when such data appears only once. Their black-box extraction method combines candidate generation with membership inference, revealing that memorization is not necessarily tied to overfitting. (Nasr et al., 2023) scale this approach to production models, introducing divergence-based attacks that extract gigabytes of training data. These methods, while powerful, often rely on access to known training data or extensive sampling infrastructure. (Jagielski et al., 2023) explore forgetting dynamics, showing that early-seen examples are more likely to be forgotten. (Meeus et al., 2024) propose copyright traps to audit memorization in models that do not naturally memorize. While membership inference attacks (MIAs) are often used to test for memorization, they are not equivalent. As (Schwarzschild et al., 2024) argue, membership is not memorization. A model may have seen a data point during training without memorizing it in a way that is problematic or reproducible. These limitations emphasize the

need for alternative approaches that directly assess memorization behavior, rather than relying solely on membership status.

**Memorization Metrics and Detection Frameworks.**   Recent work has proposed formal metrics to quantify memorization. (Carlini et al., 2023) proposed a prompting-based method to measure exact suffix reproduction, showing that memorization scales with model size and data duplication. (Zhang et al., 2023) propose counterfactual memorization, measuring the change in model predictions when specific training examples are removed. (Guo et al., 2024) use memorization signals for AI-generated text detection. (Schwarzschild et al., 2024) define the Adversarial Compression Ratio (ACR), which identifies memorized content based on the minimal prompt length required to elicit it. While effective, these approaches often require white-box access or known training data, limiting their applicability in real-world black-box settings. Our work builds on this line by introducing PEARL, a black-box framework that detects memorization via perturbation sensitivity.

## A.3   HYPERPARAMETERS

This section presents the different hyperparameter used in the different studies in Table 5.

| Param. | Description | Value |
|---|---|---|
| $k$ | Hyperparameter that allows to control the perturbation intensity. In the case of bit flips, it consists in the percentage of tokens modified in the input. | $k \in \{0, 1, 2, 3, 4, 5\}$ |
| $i$ | The number of generated outputs per perturbed input. | $i = 10$ |
| $\tau_b$ | Threshold that allows to determine whether an instance is memorized or not | Depends on the model, task and dataset considered. By default $\tau_b = 0.2$. |

Table 5: Hyperparameter description and values used during the study

## A.4   EVOLUTION OF FPR GIVEN VARIATIONS OF THE SENSITIVITY THRESHOLD $\alpha$ FOR DIFFERENT SIZES OF THE PYTHIA MODEL

To identify the relevant sensitivity thresholds for different size-variants of Pythia, we apply them on RefineWeb and assess the FPR scores that are obtained by varying the $\tau_b$ hyperparameter. The figure below provides the evolution of the FPR scores.

## A.5   EVOLUTION OF FPR ON THE LBPP DATASET GIVEN VARIATIONS OF THE SENSITIVITY THRESHOLD $\tau_b$ WITH THE GPT_4O MODEL (CODE COMPLETION TASK)

To identify an appropriate threshold for the experiment on the task dependence of IPSH, we compute the sensitivity evolution of the LBPP dataset (known to be safe) to select different values of $\tau_b$ that yield a low False Positive Rate (FPR), thereby minimizing classification errors following the completion and the summarization task.

## A.6   IPSH ILLUSTRATION EXAMPLES

This section presents the examples used to illustrate the IPSH in the main paper.

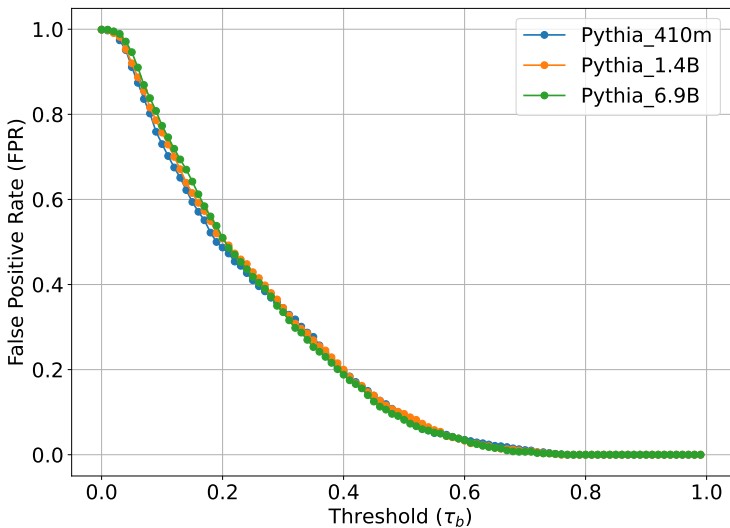

Figure 8: Evolution of FPR following the threshold $\tau_b$ regarding different model size

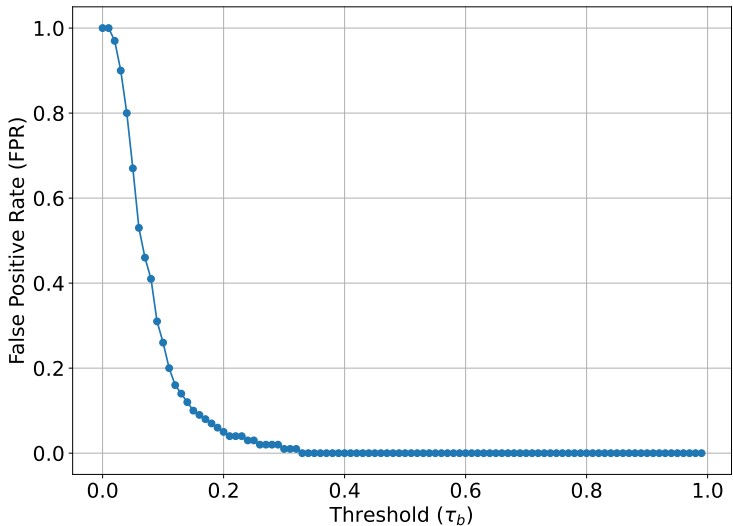

Figure 9: Evolution of FPR following the threshold (LBPP applied to GPT_4o) for completion task

### A.6.1 DATA SAMPLE IN THE TRAINING SET

**Input Shakespeare's text** : Extract of Cymbeline, King of Britain (1609).

My fault being nothing—as I have told you oft— But that two villains, whose false oaths prevail'd Before my perfect honour, swore to Cymbeline I was confederate with the Romans: so Follow'd my banishment, and this twenty years This rock and these demesnes have been my world; Where I have lived at honest freedom, paid More pious debts to heaven than in all The fore-end of my time. But up to the mountains! This is not hunters' language: he that strikes The venison first shall be the lord o' the feast; To him the other two shall minister; And we will fear no poison, which attends In place of greater state. I'll meet you in the valleys. [Exeunt GUIDERIUS and ARVIRAGUS] How hard it is to hide the sparks of nature! These boys know little they are sons to the king; Nor Cymbeline dreams that they are alive. They think they are mine; and though train'd up thus meanly I' the cave wherein they bow, their thoughts do hit The roofs of palaces, and nature prompts them In simple and low things to prince it much Beyond the trick of others. This Polydore, The heir of Cymbeline and Britain, who The king his father call'd Guiderius,—Jove! When on my three-foot stool I sit and tell The warlike feats I have done, his spirits fly out Into my story: say 'Thus, mine enemy fell, And thus I set my foot on 's neck;' even then The princely blood flows in his cheek, he sweats, Strains his young nerves and puts himself in posture That acts my words. The younger brother, Cadwal, Once Arviragus, in as like a figure, Strikes life into my speech and shows much more His own conceiving.—Hark, the [...]

**Reference output Shakespeare's text**

[...]game is roused! O Cymbeline! heaven and my conscience knows Thou didst unjustly banish me: whereon, At three and two years old, I stole these babes; Thinking to bar thee of succession, as Thou reft'st me of my lands. Euriphile, Thou wast their nurse; they took thee for their mother, And every day do honour to her grave: Myself, Belarius, that am Morgan call'd, They take for natural father. The game is up.

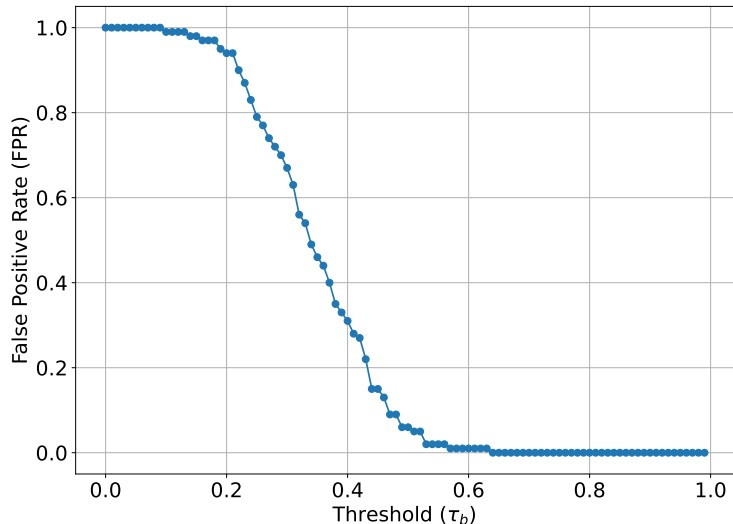

Figure 10: Evolution of FPR following the threshold (LBPP applied to GPT_4o) for summarization task

### A.6.2   DATA SAMPLE NOT IN THE TRAINING SET

> **Input BBC's article** : Extract from BBC (Published on 3rd July 2024)
>
> A humpback whale was spotted by Jersey schoolchildren returning from a day trip to Sark. Teachers and pupils from St Lawrence Primary School and d'Auvergne School captured the sighting on video and in photos as they travelled on Monday. Donna Gicquel de Gruchy, from British Divers Marine Life Rescue Channel Islands, said the humpback whale appeared to be a young one. She said it was a "very lucky and rare sighting" and she hoped it was a sign of healthy waters. Two humpback whales were also spotted off the Channel Islands in July last year. Local wildlife expert Liz Sweet described the recent sightings of the creatures in Guernsey waters as both "very rare" and "very special". Ms Sweet said the increase could be down to the whales getting confused. She said: "There is a massive migration route through the Bay of Biscay and up around the Irish coast as they head to their feeding grounds in the Arctic. "It is really busy, really noisy water and it is easy for these animals to get a little bit confused, maybe a little bit lost." Her [...]

> **Reference output BBC's article :**
>
> [...]other theory was that rising sea temperatures could be luring the whales' food closer to the Channel Islands. Ms Sweet explained: "Fish are moving around a lot more so it might have been following food so it could have just been here for a stop off.

### A.7   LIST OF IDENTIFIED GPT_4O'S MEMORIZATION INSTANCES

This section shows the data identified as memorized regarding the completion task described in the main paper with $\tau_b = 0.2$. We summarize the results in Table 6.

The instances of Bible's text that we identified as memorized are the following:

> *Memorized Instance 1:*
>
> O thou that hearest prayer, unto thee shall all flesh come.Iniquities prevail against me: as for our transgressions, thou shalt purge them away.Blessed is the man whom thou choosest, and causest to approach unto thee, that he may dwell in thy courts: we shall be satisfied with the goodness of thy house, even of thy holy temple.By terrible things in righteousness wilt thou answer us, O God of our salvation; who art the confidence of all the ends of the earth, and of them that are afar off upon the sea:Which by his strength setteth fast the mountains; being girded with power:Which stilleth the noise of the seas, the noise of their waves, and the tumult of the people.They also that dwell in the uttermost parts are afraid at thy tokens: thou makest the outgoings of the morning and evening to rejoice.Thou visitest the earth, and waterest it: thou greatly enrichest it with the river of God, which is full of water: thou preparest them corn, when thou hast so provided for it.Thou waterest the ridges thereof abundantly: thou settlest the furrows thereof: thou makest it soft with showers: thou blessest the springing thereof.Thou crownest the year with thy goodness; and thy paths drop fatness.They drop upon the pastures of the wilderness: and the little hills rejoice on every side.The pastures are clothed with flocks; the valleys also are covered over with corn; they shout for joy, they also sing.Make a joyful noise unto God, all ye lands:Sing forth the honour of his name: make his praise glorious.

| #ID | Source | $\delta(Y_0^*)$ | $\delta(Y_1^*)$ | $\delta(Y_2^*)$ | $\delta(Y_3^*)$ | $\delta(Y_4^*)$ | $\delta(Y_5^*)$ | GFD |
|-----|--------|------|------|------|------|------|------|------|
| #17 | LBPP | 0.42 | 0.23 | 0.51 | 0.18 | 0.21 | 0.25 | 0.32 |
| #58 | LBPP | 0.32 | 0.62 | 0.37 | 0.35 | 0.34 | 0.45 | 0.29 |
| #103 | HumanEval | 0.8 | 0.49 | 0.8 | 0.73 | 0.38 | 0.24 | 0.34 |
| #105 | HumanEval | 0.68 | 0.32 | 0.42 | 0.55 | 0.32 | 0.24 | 0.36 |
| #107 | HumanEval | 0.66 | 0.24 | 0.21 | 0.19 | 0.14 | 0.12 | 0.42 |
| #113 | HumanEval | 0.71 | 0.53 | 0.24 | 0.76 | 0.3 | 0.26 | 0.53 |
| #114 | HumanEval | 0.55 | 0.56 | 0.27 | 0.49 | 0.29 | 0.22 | 0.3 |
| #115 | HumanEval | 0.65 | 0.3 | 0.26 | 0.29 | 0.17 | 0.24 | 0.35 |
| #116 | HumanEval | 0.62 | 0.38 | 0.2 | 0.16 | 0.44 | 0.12 | 0.32 |
| #122 | HumanEval | 0.67 | 0.33 | 0.21 | 0.67 | 0.19 | 0.19 | 0.48 |
| #123 | HumanEval | 0.87 | 0.68 | 0.22 | 0.19 | 0.21 | 0.08 | 0.46 |
| #127 | HumanEval | 0.56 | 0.19 | 0.72 | 0.19 | 0.13 | 0.14 | 0.53 |
| #128 | HumanEval | 0.64 | 0.2 | 0.41 | 0.22 | 0.26 | 0.17 | 0.44 |
| #129 | HumanEval | 0.71 | 0.33 | 0.18 | 0.2 | 0.62 | 0.18 | 0.44 |
| #130 | HumanEval | 0.55 | 0.15 | 0.18 | 0.13 | 0.17 | 0.15 | 0.4 |
| #134 | HumanEval | 0.66 | 0.24 | 0.22 | 0.08 | 0.15 | 0.15 | 0.43 |
| #142 | HumanEval | 0.55 | 0.49 | 0.42 | 0.13 | 0.17 | 0.16 | 0.29 |
| #145 | HumanEval | 0.46 | 0.16 | 0.29 | 0.11 | 0.09 | 0.11 | 0.3 |
| #154 | HumanEval | 0.67 | 0.14 | 0.41 | 0.14 | 0.18 | 0.1 | 0.53 |
| #162 | HumanEval | 0.49 | 0.49 | 0.12 | 0.16 | 0.15 | 0.16 | 0.37 |
| #166 | HumanEval | 0.64 | 0.57 | 0.2 | 0.35 | 0.22 | 0.39 | 0.37 |
| #175 | HumanEval | 0.29 | 0.43 | 0.13 | 0.23 | 0.14 | 0.23 | 0.3 |
| #180 | HumanEval | 0.7 | 0.67 | 0.71 | 0.29 | 0.34 | 0.33 | 0.42 |
| #188 | HumanEval | 0.43 | 0.13 | 0.24 | 0.16 | 0.21 | 0.15 | 0.29 |
| #198 | HumanEval | 0.67 | 0.61 | 0.32 | 0.31 | 0.34 | 0.35 | 0.29 |
| #262 | Bible | 0.65 | 0.18 | 0.18 | 0.09 | 0.1 | 0.09 | 0.47 |
| #289 | Bible | 0.63 | 0.23 | 0.09 | 0.09 | 0.08 | 0.08 | 0.4 |

Table 6: List of identified memorized cases in GPT_4o regarding the text completion task with $\delta(Y_k^*)$ the mean of Normalized Compression Distance obtained at perturbation $k$.

---

*Memorized Instance 2:*

But let every man prove his own work, and then shall he have rejoicing in himself alone, and not in another.For every man shall bear his own burden.Let him that is taught in the word communicate unto him that teacheth in all good things.Be not deceived; God is not mocked: for whatsoever a man soweth, that shall he also reap.For he that soweth to his flesh shall of the flesh reap corruption; but he that soweth to the Spirit shall of the Spirit reap life everlasting.And let us not be weary in well doing: for in due season we shall reap, if we faint not.As we have therefore opportunity, let us do good unto all men, especially unto them who are of the household of faith.Ye see how large a letter I have written unto you with mine own hand.As many as desire to make a fair shew in the flesh, they constrain you to be circumcised; only lest they should suffer persecution for the cross of Christ.For neither they themselves who are circumcised keep the law; but desire to have you circumcised, that they may glory in your flesh.But God forbid that I should glory, save in the cross of our Lord Jesus Christ, by whom the world is crucified unto me, and I unto the world.For in Christ Jesus neither circumcision availeth any thing, nor uncircumcision, but a new creature.And as many as walk according to this rule, peace be on them, and mercy, and upon the Israel of God.

---

## A.8 EXAMPLES OF HIGH AND LOW SENSITIVITY IN REAL-WORLD TEXTS

This section presents examples of inputs from the NY Times and Bible datasets that exhibit the lowest and highest sensitivity scores, based on the completion task described in in the main paper. These examples, summarized in Table 7 and shown in and shown in Figure 11 and Figure 12, illustrate the range of model behavior under input perturbation.

The observed variation in sensitivity highlights how differently the model responds to semantically similar inputs across sources. Notably, the highest sensitivity scores suggest potential memorization, while the lowest scores indicate more stable, generalized behavior. These extremes can serve as empirical anchors for selecting the sensitivity threshold $\alpha$ in practice, helping to distinguish memorized instances from generalizable ones.

| Source | Lowest GFD | | Highest GFD | |
|--------|------|------|------|------|
| | #ID | GFD | #ID | GFD |
| Bible | #261 | 0.019 | #262 | 0.469 |
| NY Times | #301 | 0.007 | #304 | 0.266 |

Table 7: Examples of inputs with the lowest and highest sensitivity scores from two data sources. These extremes illustrate the range of model behavior under perturbation and can inform the selection of the sensitivity threshold $\alpha$ for identifying memorized instances.

> **Bible text with low sensitivity (ID : #261)**
>
> Now the sons of Reuben the firstborn of Israel, (for he was the firstborn; but forasmuch as he defiled his father's bed, his birthright was given unto the sons of Joseph the son of Israel: and the genealogy is not to be reckoned after the birthright.For Judah prevailed above his brethren, and of him came the chief ruler; but the birthright was Joseph's:)The sons, I say, of Reuben the firstborn of Israel were, Hanoch, and Pallu, Hezron, and Carmi.The sons of Joel; Shemaiah his son, Gog his son, Shimei his son,Micah his son, Reaia his son, Baal his son,Beerah his son, whom Tilgathpilneser king of Assyria carried away captive: he was prince of the Reubenites.And his brethren by their families, when the genealogy of their generations was reckoned, were the chief, Jeiel, and Zechariah,And Bela the son of Azaz, the son of Shema, the son of Joel, who dwelt in Aroer, even unto Nebo and Baalmeon:And eastward he inhabited unto the entering in of the wilderness from the river Euphrates: because their cattle were multiplied in the land of Gilead.And in the days of Saul they made war with the Hagarites, who fell by their hand: and they dwelt in their tents throughout all the east land of Gilead.And the children of Gad dwelt over against them, in the land of Bashan unto Salcah:Joel the chief, and Shapham the next, and Jaanai, and Shaphat in Bashan.And their brethren of the house of their fathers were, Michael, and Meshullam, and Sheba, and Jorai, and Jachan, and Zia, and Heber, seven.

> **Bible text with *high* sensitivity (ID : #262)**
>
> O thou that hearest prayer, unto thee shall all flesh come.Iniquities prevail against me: as for our transgressions, thou shalt purge them away.Blessed is the man whom thou choosest, and causest to approach unto thee, that he may dwell in thy courts: we shall be satisfied with the goodness of thy house, even of thy holy temple.By terrible things in righteousness wilt thou answer us, O God of our salvation; who art the confidence of all the ends of the earth, and of them that are afar off upon the sea:Which by his strength setteth fast the mountains; being girded with power:Which stilleth the noise of the seas, the noise of their waves, and the tumult of the people.They also that dwell in the uttermost parts are afraid at thy tokens: thou makest the outgoings of the morning and evening to rejoice.Thou visitest the earth, and waterest it: thou greatly enrichest it with the river of God, which is full of water: thou preparest them corn, when thou hast so provided for it.Thou waterest the ridges thereof abundantly: thou settlest the furrows thereof: thou makest it soft with showers: thou blessest the springing thereof.Thou crownest the year with thy goodness; and thy paths drop fatness.They drop upon the pastures of the wilderness: and the little hills rejoice on every side.The pastures are clothed with flocks; the valleys also are covered over with corn; they shout for joy, they also sing.Make a joyful noise unto God, all ye lands:Sing forth the honour of his name: make his praise glorious.

Figure 11: Representative Bible passages exhibiting the lowest and highest sensitivity scores.

> **NYTimes text with low sensitivity (ID : #301)**
>
> I think my credibility would be affected if I didn't ask experts for their opinion," he said. The governor also said that the discrepancy between the predictions and the actual statistics was because of the behavior of New Yorkers themselves. With some exceptions, New Yorkers have managed to follow the restrictions on movement and socializing. Dr. Deborah Birx, the White House coronavirus response coordinator, seemed to agree and congratulated Mr. Cuomo on Friday for having slowed the tide of infections in their states. "That has dramatically changed because of the impact of what the citizens of New York and New Jersey and across Connecticut and now Rhode Island are doing to really change the course of this pandemic," Dr. Birx said. It is, of course, prudent politically and for public health reasons that elected leaders over-plan, not under-plan, for disasters. During hurricanes, for instance, governors are much more likely to save their constituents' lives, and their own jobs, by ordering evacuations early rather than banking on the chance that a storm will peter out. The main objective in "flattening the curve" of the outbreak, apart from keeping people from dying, is to slow the spread enough to keep hospitals functioning. And from the start of the coronavirus emergency, Mr. Cuomo has repeatedly taken the position that he would rather be prepared for a dire scenario that never came to pass than to blithely put his faith in optimistic forecasts. Ironically, his doomsday attitude may complicate his efforts to keep the state on course as New Yorkers start to realize that the worst has not happened and eventually get itchy to go out.

> **NYTimes text with *high* sensitivity (ID : #304)**
>
> –Monkey Eats From Bird Feeder After Escaping Scottish Wildlife Park –French Farmers Block Roads Around Paris, Escalating Protests –Russian Military Plane Crashes Near Border With Ukraine –Winter Storm Makes Landings Difficult at Heathrow –U.S. Pressed Israel to Reduce Civilian Suffering in Gaza, Blinken Says Recent episodes in Europe Historic Copenhagen Stock Exchange Partly Collapses in Fire –Historic Copenhagen Stock Exchange Partly Collapses in Fire One Person Is Killed in a Cable Car Accident in Turkey –One Person Is Killed in a Cable Car Accident in Turkey German Police Stop Pro-Palestinian Conference –German Police Stop Pro-Palestinian Conference Smoke Rings Rise From Mt. Etna –Smoke Rings Rise From Mt. Etna Nightclub Fire in Istanbul Kills 29 People 0:48 Nightclub Fire in Istanbul Kills 29 People Waiters Compete in Paris' Revived Cafe Race –Waiters Compete in Paris' Revived Cafe Race Princess of Wales Announces Cancer Diagnosis –Princess of Wales Announces Cancer Diagnosis Russian Strikes Cut Off Electricity and Disrupt Water Supply in Kharkiv –Russian Strikes Cut Off Electricity and Disrupt Water Supply in Kharkiv Homes Are Destroyed by Russian Attack in Southeastern Ukraine –Homes Are Destroyed by Russian Attack in Southeastern Ukraine Missile Attack on Kyiv –Missile Attack on Kyiv Volcano Erupts in Southwestern Iceland –Volcano Erupts in Southwestern Iceland France Enshrines Abortion Rights in Its Constitution –France Enshrines Abortion Rights in Its Constitution.

Figure 12: Representative NYTimes text exhibiting the lowest and highest sensitivity scores.

## A.9  IMPACT OF THE DIFFERENT PERTURBATION ON THE INPUT

This section presents an illustration of the impact of different text perturbation strategies employed in this study, including bit-flip and synonym substitution. Table 8 demonstrates the application of these perturbations with hyperparameter $k \in \{1, 2, 3, 4, 5\}$, showing how progressive perturbation affects the original input.

## A.10  COMPARISON MEMBERSHIP VS MEMORIZATION

This section presents the illustration of the result of the comparison study between the membership inference attack using the perplexity study (Carlini et al., 2021) and the memorization using IPSH as described in the main paper. We perform this experiment using Pile (Finetuning data) and RefineWeb (Unknown data) with the different checkpoints on the Pythia model. As described in the main paper, the objective is to identify instances exhibiting low perplexity, which indicates training membership (Carlini et al., 2021). The threshold is determined by identifying the inflection point where the false positive rate (FPR) becomes sufficiently low to effectively classify potential member instances. We observe the following phenomenon, illustrated in Figure 13a.❶ The analysis confirms the hypothesized asymmetry: while memorized samples (high drop values) consistently exhibit low perplexity, many low-perplexity data points are not necessarily memorized. This is particularly

| Perturbation Type | k | Perturbed input | Cosine Distance |
|---|---|---|---|
| | 0 | In the end, we will remember not the words of our enemies, but the silence of our friends. | 0.000 |
| Bit flips | 1 | In the end, we will-based not the words of our enemies, but the silence of our friends. | 0.062 |
| | 2 | Inro end, we will remember not theMS of our enemies, but the silence of our friends. | 0.070 |
| | 3 | In the end, weull remember not the words of our enemies,é the silence of our friends. | 0.098 |
| | 4 | In the end, we will remember notas words of======== enemies, but the silence of our friends. | 0.121 |
| | 0 | In the end, we will remember not the words of our enemies, but the but the silence of our friends. | 0.000 |
| Synonym Substitution | 1 | In the end, we must remember not the words to our enemies, but the silence of our friends. | 0.075 |
| | 2 | In the end, we must remember never the words to our enemies, but the silence of friends. | 0.104 |
| | 3 | In the ending, we must remember never the words to our enemies, but the silence of our friends. | 0.109 |
| | 4 | In the ending, we must remember never the speech to our enemies, but the silence of the friends. | 0.159 |

Table 8: Illustration of the impact of the Bit flips and Synonym transformations on the input by evaluating the increase in distance between the perturbed input and the original input as a function of the hyperparameter k.

evident in later epochs, where the low-perplexity region expands significantly. ❷ The relationship between membership inference and memorization evolves during training for both datasets. The finetuning dataset shows an increasing alignment between the two detection methods as training progresses. ❸ In contrast, the patterns observed with the unknown data (RefineWeb) demonstrate a near-complete disconnect between memorization and low perplexity. The vast majority of low-perplexity samples do not meet the memorization threshold, and the high-sensitivity regions remain minimal across all epochs. This suggests that the memorization pattern described by IPSH is negligible with unseen data.

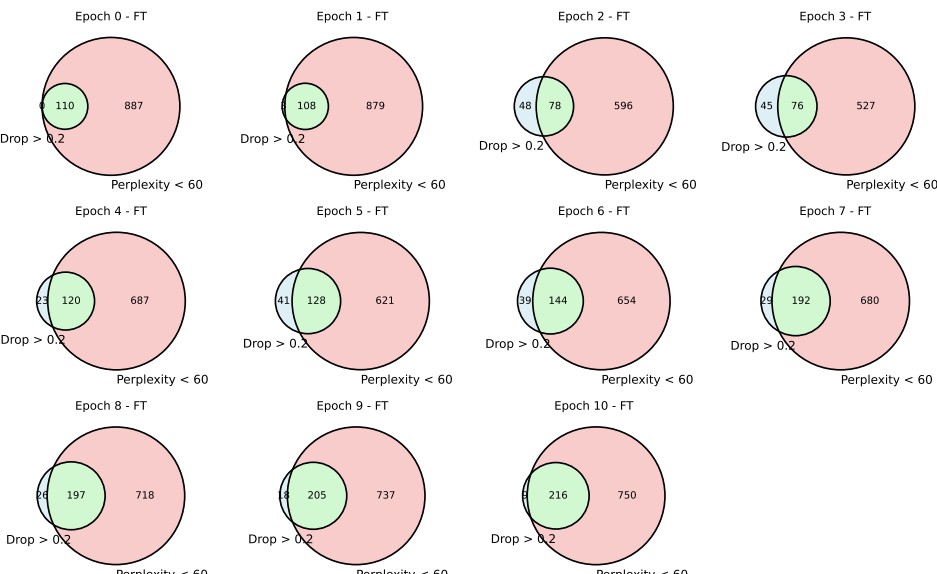

(a) Venn diagrams showing the relationship between memorization $\tau_b = 0.2$ and membership inference $Perplexity \leq 60$ for Pile (finetuned) data across training epochs. The intersection (green) represents samples satisfying both conditions, demonstrating increasing alignment between memorization and low perplexity as training progresses.

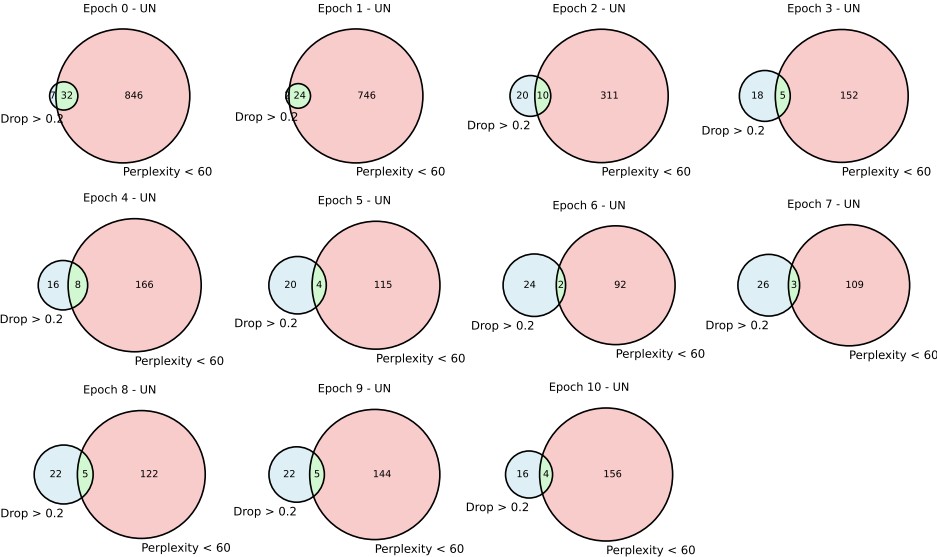

(b) Venn diagrams showing the relationship between memorization $\tau_b = 0.2$ and membership inference $Perplexity \leq 60$ for RefineWeb (unknown data) across training epochs. The minimal intersection (green) reveals that memorization and low perplexity rarely coincide for unknown data, with intersection values decreasing.

Figure 13: Evolution of memorization detection via IPSH $\alpha = 0.2$ versus membership inference via perplexity analysis $Perplexity < 60$ across different checkpoint of fine-tuned Pythia model. The analysis reveals fundamentally different behavioral patterns: ❶ fine-tuned data exhibits progressive convergence between the two detection methods, while ❷ unknown data maintains persistent divergence with intersection.

