# OpenReview forum: "Memorization or Interpolation? Detecting LLM Memorization through Input Perturbation Analysis"
_ICLR.cc/2026/Conference — Submitted to ICLR 2026_

### Official Review · Reviewer_rA7W · 2025-10-17

**Soundness:** 2
**Presentation:** 3
**Contribution:** 2
**Rating:** 2
**Confidence:** 4

**Summary:**

This paper studies the interplay of memorization and generalization in LLMs.

The paper proposes a definition of the *memorization advantage*, which is the performance gain of the model due to memorization, as opposed to "genuine" generalization.

The paper then proposes the *Input Perturbation Sensitivity Hypothesis*. The hypothesis is that the performance of a model around memorized datapoints is sensitive to perturbations.

The Input Perturbation Sensitivity Hypothesis then leads to PEARL, a method where the bits of the input are flipped, and the resulting performance drop of the model is measured against a threshold to determine memorization.

The paper then presents applications of PEARL to the Bible, HumaEval, and New York Times articles for Pythia models and GPT-4o.

**Strengths:**

- This paper examines an interesting and challenging question that is highly relevant to the ICLR community.

- The paper is overall really well-written and good to follow.

- The proposed definition of the memorization advantage is interesting.

- The idea of using input perturbations, while not novel, is good. It is worth studying this approach.

- I appreciate that the authors provide an anonymous code repository.

**Weaknesses:**

**The idea of using input perturbations to identify localized memorization and contamination in LLMs is not really novel.** Variants of this idea have been explored in previous works. For example:

"GSM-Symbolic: Understanding the Limitations of Mathematical Reasoning in Large Language Models", https://arxiv.org/abs/2410.05229

"Elephants Never Forget: Memorization and Learning of Tabular Data in Large Language Models", https://arxiv.org/pdf/2404.06209

"Rethinking Benchmark and Contamination for Language Models with Rephrased Samples", https://arxiv.org/pdf/2311.04850

"Data-Copying in Generative Models: A Formal Framework", https://arxiv.org/abs/2302.13181

**Refined web is claimed not to be included in the Pile, without evidence.** You write that "A key step is determining a sensitivity threshold τ_b for identifying memorized instances. We calibrate this threshold on the RefinedWeb dataset, which is known to be excluded from Pythia’s training data." (L293)

Now, refined web is based on the Common Crawl (https://arxiv.org/pdf/2306.01116)

The Pile, Pythia’s training data, is also based on Common Crawl and other internet data (https://arxiv.org/abs/2101.00027)

How can Refined web be "know to be excluded" from the Pile? The calibration of the threshold is a crucial step, this seems like a **major flaw in the analysis.**

**The difficulty of naturally perturbing news articles and books is not discussed.** The paper proposes to use low-level bit flips to measure memorization (PEARL). However, observed performance drops after such bit flips can be due to many reasons, not necessarily memorization. This is especially the case for natural text, where bit flips might lead to samples outside the data distribution. This seems like a crucial point that is not discussed in the paper.

**The memorization analysis of New York Times articles is flawed.** The problem is the following: After the New York Times lawsuit, OpenAI, with high likelihood, inserted safeguards into their system that prohibited the model from sending responses that closely correspond to New York Times articles. This effect was actually observable around January 2024, when the extraction attacks listed in the New York Times lawsuit suddenly stopped working. So here you are not really analyzing the LLM itself, but rather the behavior of the safeguards that OpenAI has built around their LLMs.

**Many of the observations around the interplay between memorization and generalization are subjective and not appropriately supported with evidence.** The interplay between memorization and generalization in LLMs is a surprisingly subtle topic. The authors of this paper have clearly thought about this topic, but the analysis is not yet rigorous enough, and their intuitions are not sufficiently supported with evidence. To give just one example, the authors write that

*"While some argue memorization may precede generalization (Feldman, 2020; Dankers & Titov, 2024), it ultimately signifies fragile performance, a key indicator of overfitting (Schwarzschild et al., 2024; Duan et al., 2024)."* (L: 146)

I would argue that, actually, we have very little idea whether this is the case or not. Indeed, there is extensive work on benign overfitting in deep learning theory (https://arxiv.org/abs/1906.11300). So why is it clear that LLMs cannot generalize and interpolate training data points simultaneously?

**Questions:**

**Q1:** You argue that prior work that demonstrates memorization in black-box LLMs relies on the fact that we already know parts of the training data (e.g. "Scalable Extraction of Training Data from (Production) Language Models", https://arxiv.org/pdf/2311.17035). I certainly agree with this point, and that it can be seen as a limitation of existing methods. However, it seems to me that your method at least partly relies on the assumption that we have data that is similar (even iid. ?) to the training data, but where we know that it was not included into training - I believe that you need this for the calibration of the capability threshold, see also the point above. Now: Is this not an equally strong, if not a much stronger requirement?

**Q2:** The definition of memorization. This is an understanding question. When you say "memorization" in the context of LLMs, is there any particular pre-existing definition of this term that you rely on, independently of the definitions that are introduced in this paper? Or are you using the term "memorization" intuitively, proposing new definitions, that is independently of existing definitions like extractable memorization?

**Further comments:**

- I would suggest including the formal statement of the Hypothesis (IPSH) from L667 in the Supplement into the main paper.

- L257: "We thus proposed to investigate" (Typo)

**Justification for final score**:

This paper studies an interesting question and is well-written. The authors clearly invested work into this paper. The introduction of the paper was interesting to read, and there are many interesting thoughts about memorization and generalization in this paper.

The main idea of using input perturbations is not as novel as claimed by the authors, but this would be ok if there was interesting new analysis.

Unfortunately, the current version of the paper overall lacks the rigor that would be expected from a conference paper at ICLR. There are a couple of points in the analysis that I believe are fundamentally flawed, including the claim that Refined web is not included in the Pile, and measuring memorized NYT articles in gpt-4o. In addition, many of the observations around memorization and generalization, while interesting, are ultimately subjective and not sufficiently supported by evidence. For these reasons, I suggest to reject this paper at this point.

---

### Official Review · Reviewer_pwqr · 2025-10-26

**Soundness:** 1
**Presentation:** 2
**Contribution:** 2
**Rating:** 2
**Confidence:** 3

**Summary:**

The paper proposes to detect memorization based on input perturbation analysis. The principle hypothesis is that, for a given model, task, and data point, if the model has memorized the data point, then its task performance will be sensitive to perturbations of that data point. Perturbation is usually performed on the initial part of the data point, called the prompt, and the authors check if the suffix is very different and low performing compared to reference suffix (or later part of the data point).

**Strengths:**

The proposed method detects memorization without requiring access to training data or model internals. Experiments are conducted on both open-source and closed-source models. Memorization is reported from diverse domains, including Bible texts, benchmark code, and proprietary contents.

**Weaknesses:**

I have a critical opinion about the soundness of the proposed idea.

- The hypothesis of relying on input perturbation to decide memorization based on performance degradation is subjective, and I do not agree with the hypothesis. Referring to table 1, the example of memorization (2nd row) is not a valid example. The model already identifies spelling mistakes and missing numbers. I am not sure how this is an indication of memorization, where the model clearly identifies that the input prompt is substandard (in fact, it looks like generalization). Also, generalization and memorization are picked from different sources -- what would be an example of generalization in the context of Harry Potter books? In fact, I would argue that if the model still generates the same/similar response even if input perturbation is applied, then it is a case of memorization, contradicting the original hypothesis. Because, in this case, the model regurgitates the response even if the input prompt is slightly different. Also, the hypothesis in the introduction is not proved rigorously in later part of the paper.

- The memorization examples in Table 1 (the input part) do not satisfy the semantically equivalent criteria mentioned in Line 185.

- The proposed method relies on several thresholds to decide memorization. The subjectivity of threshold has influences on when a particular data point is treated as memorized or not. I suggest a comparison with [1], where memorization is clearly disentangled from generalization (or learning), and the subjectivity of choosing thresholds is avoided.

- Equation (1) is not explained well. How do you find $D_{\text{test}}$ and $D'_{\text{test}}$?

[1] Rethinking Memorization Measures in LLMs: Recollection vs. Counterfactual vs. Contextual Memorization

**Questions:**

Please address the weakness part above.

---

### Official Review · Reviewer_LU9B · 2025-10-31

**Soundness:** 2
**Presentation:** 3
**Contribution:** 2
**Rating:** 2
**Confidence:** 3

**Summary:**

The paper introduces PEARL, a new method to test LLM memorization in black-box settings. The method works by looking at changes in the output w.r.t. changes in the input and hinges on the hypothesis that memorized samples show more significant changes in this setup due to the model not generalizing to them. The perturbations are performed using bit flips as well as synonym replacements and the authors test on both an open-source model family (Pythia) and a closed source model (GPT-4o). The authors find that this approach is able to capture several cases of memorization in both setups and hence can be used to detect the same in the wild.

**Strengths:**

- The problem under study is timely
- The authors test on both open as well as closed models
- The authors explore several angles (such as task dependence, relation to membership inference, impact of model size)

**Weaknesses:**

- I think the idea itself is not novel or new. Similar ideas have existed for a while (See [1] and [2] where similar ideas are used). The connection being used by the authors is that memorization leads to overfitting to a certain datapoint and hence the model will do much more poorly on nearby datapoints.
- I am not convinced that bitflip is a good mechanism because it can change the language or even potentially transform to a non-existent UTF-8 ID. In such a situation where you have very weird characters in the input, I won't be surprised if the model does not perform similarly as the original setting.
- For the Pythia analysis, by the very act of first instilling memorization aren't you weakening the study? This is no longer a study of memorization on pre-trained models but one on model which are forced to memorize certain content and then the method is evaluated on the very same content? Unless I'm missing something there is no discussion on detecting memorization of content which is seen during pretraining only.
- The paper has a bunch of unnecessary figures which are just serving as distractors (e.g. Fig 5, 7). These figures can easily be mentioned in text.


[1] Detecting Overfitting via Adversarial Examples (https://arxiv.org/abs/1903.02380)
[2] Model Validation Using Mutated Training Labels: An Exploratory Study (https://arxiv.org/abs/1905.10201)

**Questions:**

Question -

How do you ensure the samples from RefinedWeb are disjoint from Pile? Don't they share a lot of sources. Can you cite something for the claim in Lines 294-296?

Nit -

Line 77-78 - Brackets not closed

---

### Official Review · Reviewer_fVik · 2025-11-01

**Soundness:** 2
**Presentation:** 3
**Contribution:** 2
**Rating:** 2
**Confidence:** 4

**Summary:**

This work aims to detect memorization in a blackbox setting, without access to training data. The authors propose a method based on the hypothesis that memorized examples will be more sensitive to input perturbations, which measures how much does model prediction change with small semantic-preserving perturbations. The authors first validate their methods in a controlled setting, and then experiment it on GPT-4 over four data sources that potentially have different levels of memorization. Overall, the authors conclude that sensitivity to input perturbations provide a good measurement of memorization.

**Strengths:**

* This work aims to solve a very ambitious problem, how to detect memorization (1) without access to training data (2) without access to model internals.
* The paper is generally easy to follow.

**Weaknesses:**

* **The idea of using perturbations to detect memorization has been explored in the literature**: While the authors present the input perturbation sensitivity hypothesis as a main contribution, this idea is not new to the community, especially in the area of detecting dataset contamination. For example, [Oren et al. 2023](https://openreview.net/pdf?id=KS8mIvetg2) proposed a permutation test that not only detects dataset contamination, but statistically quantify the uncertainty. Similarly, [Mirzadeh et al. 2024](https://openreview.net/forum?id=AjXkRZIvjB) perturbed the GSM-8K to distinguish memorization from reasoning. It would be great if the authors can connect with this line of work in revisions.

* **The input perturbation sensitivity hypothesis is validated under an unrealistic setting that might cause overfitting**: In the validation setting, the authors fine-tuned the pythia models on the positive set (1000 sequences) for 10 epochs to induce memorization -- this unfortunately might just overfit the pre-trained model on the positive set and exaggerate the memorization effects. In contrast, sequences naturally memorized through pre-training are exposed at a much lower frequency, allowing the model to preserve general knowledge. The authors might want to redo this experiment by using sequences verbatim memorized by pre-trained pythia models. There are plenty of such sequences.

* **The results on GPT-4 is a bit preliminary**: The test is only run on four sources, each with 100 sequences, and we only have speculative labels on which sources are memorized. Moreover, out of the four sources, Bible and HumanEval have a ranking that is even flipped from the speculative labels. This unfortunately makes it extremely hard to tell if the method is actually working or not. Maybe the authors can run more experiments on dataset/models where we have access to ground truth before doing speculative experiment on GPT-4.

**Questions:**

* Could the authors clarify what "memorized the data point" mean on line 75?
* The bit flip strategy in neighbor data point generation seems ad-hoc. What's an intuition that this strategy would work? Is this similar to introducing typos inputs?
* Is the threshold 0.2 on line 323 a typo?

---

### Author Response · Authors · 2025-11-30

We thank the reviewers for taking the time to review our paper and provide detailed feedback. We appreciate the effort and insights shared and will carefully consider all comments and suggestions in future iterations of this work. We would like to clarify a key point: the core contribution of our paper is not merely perturbation. Rather, we formalize PEARL — a methodological framework for detecting memorization in machine learning models. Memorization detection is inherently ambiguous, as existing approaches often lack a clear operational definition. Our work addresses this gap by providing a principled way to measure memorization via input perturbation analysis, which compares instance performance under controlled perturbations against calibrated baselines. This operationalization is crucial for moving beyond intuition toward a rigorous, empirically grounded detection system built on established learning theory principles.

---

### Meta-Review · Area_Chair_UFCW · 2026-01-06

**Summary:**

This submission studies the important and timely problem of detecting memorization in large language models under black-box access. The authors propose PEARL, a framework based on the Input Perturbation Sensitivity Hypothesis, and evaluate it on both open-source and closed-source models.

However, there is strong consensus across the reviews that the paper does not yet meet the acceptance bar for ICLR. While the problem itself is highly relevant, the core methodological idea is not considered novel. Closely related perturbation- or mutation-based approaches have been explored in prior work, and the paper does not sufficiently situate PEARL within this literature or clearly articulate what fundamentally new capability it offers beyond existing methods.

Reviewers also raise significant concerns regarding the soundness and rigor of the experimental validation. The calibration procedure relies on the assumption that RefinedWeb is excluded from the Pile, which is not convincingly justified. The use of low-level bit flips as perturbations is potentially distribution-shifting, making it difficult to attribute performance degradation specifically to memorization rather than input corruption. Possible confounding effects from system-level safeguards (e.g., for New York Times content) complicate the interpretation of the closed-model results. More broadly, the central hypothesis linking perturbation sensitivity to memorization is not rigorously established nor clearly disentangled from generalization behavior.

In summary, while the paper is clearly written and addresses an important problem, the lack of novelty, limited engagement with prior work, and concerns about experimental design and interpretability substantially limit its contribution in its current form. On balance, I recommend rejection at this time.

**Reviewer Concerns:**

The authors did not provide a rebuttal.

**Reviewer Scores:**

The authors did not provide a rebuttal

---

### Decision · Program_Chairs · 2026-01-26

Reject